# On the Evaluation of Higher-Harmonic-Current Responses for High-Field Spectroscopies in Disordered Superconductors

Götz Seibold

Institut für Physik, BTU Cottbus-Senftenberg, D-03013 Cottbus, Germany; seibold@b-tu.de

**Abstract:** We discuss a formalism that allows for the calculation of a higher-harmonic-current response to a strong applied electric field for disordered superconducting systems described on the basis of tight-binding models with on- and/or intersite interactions. The theory is based on an expansion of the density matrix in powers of the field amplitudes, where we solve the equation of motion for the individual components. This allows the evaluation of higher-order response functions on significantly larger lattices than one can achieve with a previously used approach, which is based on a direct temporal integration of the equation of motion for the complete density matrix. In the case of small lattices, where both methods can be applied by including also the contribution of collective modes, we demonstrate the agreement of the corresponding results.

**Keywords:** superconductivity; third-harmonic generation; disorder

## 1. Introduction

Both linear and non-linear response spectroscopies provide valuable and complementary information on the excitations of high-temperature superconductors. Since the discovery of these materials, optical conductivity measurements have been central in advancing our understanding of the unusual electronic properties, including, e.g., the superconducting gap, the pseudogap, or the transition from a Mott-insulating state to a (non-)Fermi liquid (for a review, see, e.g., [1]).

On the other hand, the development of non-equilibrium spectroscopies has significantly advanced our understanding of complex materials, due to the possibility of disentangling different dynamical processes via their different relaxation times [2]. With regard to superconducting materials, measurements of the non-linear current response have been recently been applied in order to elucidate the order parameter dynamics, which, as a scalar quantity, is not visible in the equilibrium response. Corresponding experiments have been conducted in NbN [3–5], MgB$_2$ [6,7], pnictides [8], and cuprate superconductors [9–12], whereas the theoretical understanding of these studies was advanced in [13–24]. Basically, the current density in response to an applied vector potential $\mathbf{A}(t)$ can be expanded up to third order as $j_\alpha = \chi^{(1)}_{\alpha\beta} A_\beta + \chi^{(3)}_{\alpha\beta\gamma\delta} A_\beta A_\gamma A_\delta$, where $\chi^{(1)}$ is the linear response, which is related to the optical conductivity. On the other hand, $\chi^{(3)}$ incorporates processes where the (scalar) order parameter fluctuations $\delta\Delta$ are driven by terms quadratic in $\mathbf{A}(t)$ so that the third harmonic generation (THG) is expected to be enhanced at twice the frequency of the incoming field $2\omega$ corresponding to the spectral gap $2\Delta$ of the superconductor (SC). However, it has been shown [13] that, for clean single-band s-wave SCs, these amplitude ("Higgs") excitations yield only a minor contribution to the THG, which instead is dominated by the BCS quasiparticle excitations across the SC spectral gap. For a square lattice, the amplitude excitations only become visible when the THG is measured at an angle of $\pi/4$ with respect to the bond direction, which suppresses the QP contribution. For a clean system, the response is only due to the diamagnetic current, while disorder induces also a paramagnetic contribution [4,15,16,19]. It has been shown [19] that, at moderate

disorder, the response is still dominated by the BCS part, while collective modes may yield comparable contributions only in the limit of strong disorder.

In this context, various approximations for the theoretical description of disorder within the BCS theory have been considered. In the weakly disordered limit $k_F l \gg 1$, previous work [16,21] was either based on the Mattis–Bardeen model [25] or on the self-consistent Born approximation [4]. The summation of diagrams with impurity ladders, equivalent to the solution of quasiclassical Eilenberger equations and formally valid for arbitrary scattering rate, was accomplished in [15]. In our previous work [19,26], we treated the influence of disorder on the THG *exactly* by solving the time-dependent Bogoljubov–de Gennes equations on finite lattices with local Anderson-type disorder. Formally, this has been achieved by adding a time-dependent vector potential to the Hamiltonian and by computing the resulting dynamics from the equation of motion for the time-dependent density matrix. This formalism can be accomplished in two different ways, which have been used in [19] and [26], respectively. First, the dynamics of the full density matrix can be computed from the equation of motion, and at the end, the various harmonic contributions, proportional to the corresponding power in the amplitude of the applied vector potential $\sim A_0^n$, have to be extracted numerically; see [19]. This is a rather flexible approach, which allows considering the influence of collective modes (amplitude, phase, charge) and, in principle, also allows incorporating different pump–probe protocols. However, for a lattice with $N$ sites, the density matrix for a superconductor has dimensions $2N \times 2N$ so that the formalism is restricted to small lattices only. Second, as outlined in [26], the density matrix can be expanded from the beginning in powers of the applied vector potential. The equations of motion can be immediately formulated in frequency space for the individual components and allow for the computation of the current response at the respective order. This approach is less flexible in the time domain, but can be applied to much larger lattices as relevant for the investigation of d-wave superconductors, at least on the BCS level, as was demonstrated in [26].

Here, we compared in detail both formalisms and also discuss how collective modes can be incorporated into the second approach. Section 2 introduces the model, and we will analyse the two different approaches for the computation of the THG in a disordered tight-binding lattice with attractive on-site interaction ("attractive Hubbard model") in Section 3. In the same section, we also compare the outcome of both procedures for the case of a disordered s-wave system. We conclude our discussion in Section 4.

## 2. Model

We illustrate our formalism within the attractive Hubbard model on a square lattice plus local on-site disorder (cf., e.g., [19,27–29]):

$$H = \sum_{ij\sigma} t_{ij} c_{i\sigma}^{\dagger} c_{j\sigma} - |U| \sum_i n_{i\uparrow} n_{i\downarrow} + \sum_{i\sigma} V_i n_{i\sigma} \tag{1}$$

where the local potential $V_i$ is taken from a flat distribution $-V_0 \le V_i \le +V_0$.

To describe the SC state, Equation (1) is solved in the mean-field using the Bogoljubov-de-Gennes (BdG) transformation:

$$c_{i\sigma} = \sum_k \left[ u_i(k)\gamma_{k,\sigma} - \sigma v_i^*(k)\gamma_{k,-\sigma}^{\dagger} \right]$$

which yields the eigenvalue equations:

$$\omega_k u_n(k) \quad = \sum_j t_{nj} u_j(k) + [V_n - \tfrac{|U|}{2}\langle n_n \rangle - \mu] u_n(k) + \Delta_n v_n(k) \tag{2}$$

$$\omega_k v_n(k) \quad = -\sum_j t_{nj}^* v_j(k) - [V_n - \tfrac{|U|}{2}\langle n_n \rangle - \mu] u_n(k) + \Delta_n^* u_n(k) \tag{3}$$

and the SC order parameter is defined as $\Delta_n = -|U|\langle c_{n,\downarrow} c_{n,\uparrow} \rangle$.

From the eigenvalue problem, Equations (2) and (3), one can iteratively determine the ground state density matrix $\mathcal{R}$ with the elements:

$$
\begin{aligned}
\rho_{ij} &= \langle c_{i,\uparrow}^{\dagger} c_{j,\uparrow} \rangle = \sum_{k} \left[ v_i(k) v_j^*(k)(1 - f(E_k)) + u_i^*(k) u_j(k) f(E_k) \right] \\
\bar{\rho}_{ij} &= \langle c_{i,\downarrow} c_{j,\downarrow}^{\dagger} \rangle = \sum_{k} \left[ u_i(k) u_j^*(k)(1 - f(E_k)) + v_i^*(k) v_j(k) f(E_k) \right] \\
\kappa_{ij} &= \langle c_{i,\downarrow} c_{j,\uparrow} \rangle = \sum_{k} \left[ -u_i(k) v_j^*(k)(1 - f(E_k)) + v_i^*(k) u_j(k) f(E_k) \right]
\end{aligned}
$$

which in compact notation can be written as

$$
\mathcal{R} = \begin{pmatrix} \rho & \kappa^{\dagger} \\ \kappa & \bar{\rho} \end{pmatrix}.
$$

The BdG approximated energy can then be expressed via the density matrix as

$$
E^{BdG} = \sum_{ij} t_{ij} (\rho_{ij} - \bar{\rho}_{ij}) + U \sum_{i} (\rho_{ii}(1 - \bar{\rho}_{ii}) + \kappa_{ii}^* \kappa_{ii}) + \sum_{i} V_i [\rho_{ii} - \bar{\rho}_{ii} + 1],
$$

and the BdG-Hamiltonian matrix is defined as

$$
\mathcal{H}_{ij}^{BdG} = \frac{\partial E^{BdG}}{\partial \mathcal{R}_{ji}}. \tag{4}
$$

In the absence of an external field, the density matrix $\mathcal{R}$ and the Hamiltonian $\mathcal{H}^{BdG}$ commute, so that the density matrix has no time evolution. The dynamics of $\mathcal{R}(t)$ is induced via the coupling to the electromagnetic field $\vec{E}(t) = -\partial \vec{A}(t)/\partial t$. Let us consider, e.g., the case of a (spatially constant) field along the $x$ direction. $A_x(t)$ is coupled to the system via the Peierls substitution $c_{i+x,\sigma}^{\dagger} c_{i,\sigma} \rightarrow e^{iA_x(t)} c_{i+x,\sigma}^{\dagger} c_{i,\sigma}$, where, for simplicity, we will drop from the equations all the constants by putting the lattice spacing, the electronic charge $e$, the light velocity $c$, and the Planck constant $\hbar$ equal to one. The Peierls substitution modifies the kinetic energy part, leading to the following contribution to $E^{BdG}$:

$$
\begin{aligned}
T^{BdG} &= -t \left\{ e^{iA_x} \rho_{i+x,i} + e^{-iA_x} \rho_{i-x,i} - e^{-iA_x} \bar{\rho}_{i+x,i} - e^{iA_x} \bar{\rho}_{i+x,i} \right\} \\
&\quad - t' \left\{ e^{iA_x} \rho_{i+x,i+y} + e^{-iA_x} \rho_{i-x-y,i} - -e^{-iA_x} \bar{\rho}_{i+x+y,i} - e^{iA_x} \bar{\rho}_{i+x+y,i} \right. \\
&\quad + \left. e^{iA_x} \rho_{i+x,i-y} + e^{-iA_x} \rho_{i-x+y,i} - e^{-iA_x} \bar{\rho}_{i+x-y,i} - e^{iA_x} \bar{\rho}_{i+x-y,i} \right\}
\end{aligned} \tag{5}
$$

where we included a nearest ($\sim t$) and next-nearest ($\sim t'$) neighbour hopping into the Hamiltonian.

## 3. Computation of the Dynamics

The equation of motion for the density matrix reads

$$
i \frac{d}{dt} \mathcal{R} = \left[ \mathcal{R}, \mathcal{H}^{BdG} \right] \tag{6}
$$

with the BdG-Hamiltonian matrix given by Equation (4).

Solving Equation (6) yields a time-dependent BdG energy $E^{BdG}(t)$ and, thus, a time-dependent current density, which is obtained from

$$
j_x(t) = -\frac{1}{N} \frac{\partial E^{BdG}}{\partial A_x} = -\frac{1}{N} \frac{\partial T^{BdG}(t)}{\partial A_x} \tag{7}
$$

with $N$ denoting the number of sites. The task is now to evaluate the current response for a given order in the amplitude of the applied vector potential. As a first step, we expand Equation (5) up to third order in $A_x$, which yields

$$j_x = \left(1 - \frac{1}{2}A_x^2\right)j_{para}^x + A_x\left(1 - \frac{1}{6}A_x^2\right)j_{dia}^x \tag{8}$$

with

$$
\begin{aligned}
j_{para}^x &= it\sum_n[\rho_{n+x,n} - \bar\rho_{n-x,n} - \rho_{n-x,n} + \bar\rho_{n+x,n}] \\
&+ it'\sum_n[\rho_{n+x+y,n} - \bar\rho_{n-x-y,n} - \rho_{n-x-y,n} + \bar\rho_{n+x+y,n}] \\
&+ it'\sum_n[\rho_{n+x-y,n} - \bar\rho_{n-x+y,n} - \rho_{n-x+y,n} + \bar\rho_{n+x-y,n}] \\
j_{dia}^x &= -t\sum_n[\rho_{n+x,n} - \bar\rho_{n-x,n} + \rho_{n-x,n} - \bar\rho_{n+x,n}] \\
&- t'\sum_n[\rho_{n+x+y,n} - \bar\rho_{n-x-y,n} + \rho_{n-x-y,n} - \bar\rho_{n+x+y,n}] \\
&- t'\sum_n[\rho_{n+x-y,n} - \bar\rho_{n-x+y,n} + \rho_{n-x+y,n} - \bar\rho_{n+x-y,n}].
\end{aligned}
$$

Here, the subscripts *para* and *dia* refer to the usual identification of the leading terms coupling the gauge field to the fermionic operators in the Hamiltonian, i.e., the linear coupling between the paramagnetic term and $A_x$ and a quadratic coupling between the electronic density and $A_x^2$, which leads to the standard diamagnetic contribution to the current in the linear response. However, both $j_{para}^x$ and $j_{dia}^x$ still contain the vector potential to all orders in $A_x$.

Writing $A_x(t) = A_0 f(t)$, we can expand the currents in a power series in $A_0$:

$$
\begin{aligned}
j_{para}^x &= \sum_n A_0^n j_{para}^{x,(n)} \\
j_{dia}^x &= \sum_n A_0^n j_{dia}^{x,(n)}
\end{aligned}
$$

which, upon inserting into Equation (8), allows us to extract the various current contributions to order $n$, $j_x^{(n)}$. In particular, the third harmonic contribution to the current density reads

$$j_x^{(3)}(t) = j_{para}^{x,(3)}(t) - \frac{1}{2}f^2(t)j_{para}^{x,(1)}(t) + f(t)j_{dia}^{x,(2)}(t) - \frac{1}{6}f^3(t)j_{dia}^{x,(0)} \tag{9}$$

where we find that the dominant paramagnetic and diamagnetic contributions are given by $j_{para}^{x,(3)}$ and $A_0 j_{dia}^{x,(2)}$. On the other hand, $j_{para}^{x,(1)}$ and $j_{dia}^{x,(0)}$ also enter the calculation of the optical conductivity in first order.

In [19], we numerically integrated Equation (6) using an Adams–Bashforth algorithm and an initialising fourth-order Runge–Kutta method. The resulting time-dependent currents $j_{para}^x$ and $j_{dia}^x$ then were separated numerically into the individual components $j_{para}^{x,(n)}$ and $j_{dia}^{x,(n)}$ from which, after the Fourier transformation, the frequency-dependent third harmonic response Equation (9) was evaluated. In particular, at low energies, this procedure is rather time consuming since the integration has to be performed over several periods of the incoming field.

Here, we compared this approach with a different strategy, where from the beginning, we expanded the density matrix in powers of the applied vector potential:

$$\mathcal{R} = \sum_{n=0} A_0^n \mathcal{R}^{(n)}. \tag{10}$$

Here, $\mathcal{R}^{(0)}$ is the equilibrium density matrix for which

$$\left[\mathcal{R}^{(0)}, \mathcal{H}^{BdG}\right] = 0, \tag{11}$$

as we already emphasised above.

According to Equation (9), higher-order contributions to the density matrix $\mathcal{R}^{(n)}$ allow for the computation of the non-harmonic current responses $j_{para}^{x,(n)}$ and $j_{dia}^{x,(n)}$, which, as we will show in the following, can be directly obtained in the frequency space. The next subsections will address in detail the evaluation of the current responses up to third order, including the contribution from collective mode via the random phase approximation (RPA).

### 3.1. First Order

The first-order current contribution, relevant for the evaluation of the optical conductivity, is given by

$$j_x^{(1)} = j_{para}^{x,(1)} + A_0 j_{dia}^{x,(0)} \tag{12}$$

which requires the evaluation of the density matrix up to order $n = 1$.

By selecting all terms $\sim A_0$ in the equation of motion, Equation (6), one obtains

$$i\underline{\dot{R}}^{(1)} = \left[\underline{R}^{(1)}, \underline{H}^{(0)}\right] - |U|\left[\underline{R}^{(0)}, \underline{D}^{(1)}\right] + f(t)\left[\underline{R}^{(0)}, \underline{V}\right] \tag{13}$$

with

$$\underline{V} = \begin{pmatrix} \underline{v} & \underline{0} \\ \underline{0} & \underline{v} \end{pmatrix} \tag{14}$$

and

$$v_{nm} = -it[\delta_{m,n+x} - \delta_{m,n-x}] - it'\left[\delta_{m,n+x+y} - \delta_{m,n-x-y}\right] - it'\left[\delta_{m,n+x-y} - \delta_{m,n-x+y}\right].$$

The matrix $\underline{D}^{(1)}$ is defined as

$$\underline{D}^{(1)} = \begin{pmatrix} -\bar{\rho}_D^{(1)} & \kappa_D^{\dagger,(1)} \\ \kappa_D^{(1)} & -\rho_D^{(1)} \end{pmatrix} \tag{15}$$

and the subscript $D$ indicates that it only contains the diagonal elements of the respective matrices, e.g., $[\rho_D^{(1)}]_{nm} \equiv [\rho^{(1)}]_{nm}\delta_{nm}$, which are part of $\underline{R}^{(1)}$.

The non-perturbed Hamiltonian $\underline{H}^{(0)}$ (i.e., for $A_0 = 0$) can be diagonalised:

$$\underline{\tilde{H}}^{(0)} = \underline{T}^{-1}\underline{H}^{(0)}\underline{T} = \begin{pmatrix} -E_N & \dots & 0 & 0 & \dots & 0 \\ \vdots & \ddots & 0 & \vdots & \dots & \vdots \\ 0 & \dots & -E_1 & 0 & \dots & 0 \\ 0 & \dots & 0 & E_1 & \dots & 0 \\ \vdots & & \vdots & \vdots & \ddots & \vdots \\ 0 & \dots & 0 & 0 & \dots & E_N \end{pmatrix}$$

and the same transformation also diagonalises the non-perturbed density matrix:

$$
\underline{\tilde{\underline{R}}}^{(0)} = \underline{\underline{T}}^{-1}\underline{\underline{R}}^{(0)}\underline{\underline{T}} =
\begin{pmatrix}
1 & \dots & 0 & 0 & \dots & 0 \\
\vdots & \ddots & 0 & \vdots & \dots & \vdots \\
0 & \dots & 1 & 0 & \dots & 0 \\
0 & \dots & 0 & 0 & \dots & 0 \\
\vdots & & \vdots & \vdots & \ddots & \vdots \\
0 & \dots & 0 & 0 & \dots & 0
\end{pmatrix}.
$$

With this transformation, Equation (13) can be written as

$$
i\dot{\tilde{R}}^{(1)}_{nm} = (E_{mm} - E_{nn})\tilde{R}^1_{nm} - |U|(\tilde{R}^0_{nn} - \tilde{R}^0_{mm})\tilde{D}^{(1)}_{nm} + (\tilde{R}^0_{nn} - \tilde{R}^0_{mm})\tilde{V}_{nm}f(t) \tag{16}
$$

where $\underline{\tilde{\underline{V}}}$ and $\underline{\tilde{\underline{D}}}^{(1)}$ denote the transformed matrices, Equations (14) and (15).

We now perform a Fourier transformation:

$$
\tilde{R}^1_{nm}(\omega) = \int dt\, e^{i\omega t}\tilde{R}^1_{nm}(t)
$$

$$
f(\omega) = \int dt\, e^{i\omega t}f(t)
$$

so that Equation (16) reads

$$
\begin{aligned}
\tilde{R}^1_{nm}(\omega) &= \frac{\tilde{R}^0_{nn} - \tilde{R}^0_{mm}}{\omega - E_{mm} + E_{nn}}\tilde{V}_{nm}f(\omega) - |U|\frac{\tilde{R}^0_{nn} - \tilde{R}^0_{mm}}{\omega - E_{mm} + E_{nn}}\tilde{D}^{(1)}_{nm} \\
&\equiv \tilde{\chi}^{(0)}_{nm}(\omega)\tilde{V}_{nm}f(\omega) - |U|\tilde{\chi}^{(0)}_{nm}(\omega)\tilde{D}^{(1)}_{nm}.
\end{aligned} \tag{17}
$$

On the BCS level ($U = 0$), the density matrix is now obtained by transforming back to $R^{(1)}_{ij}$ in the original site representation. For the practical computation, $\tilde{\chi}_{nm}(\omega \to \omega - i\epsilon)$ should be shifted into the complex plane in order to avoid singularities.

Including fluctuations means including the corrections due to the matrix $\underline{\underline{D}}^{(1)}$. In the original site representation and in the case of local interactions (as in the present case of the attractive Hubbard model), $\underline{\underline{D}}^{(1)}$ has only diagonal elements in $\rho$ and $\kappa$, which in the following, we denote by Greek letters, i.e., $D^{(1)}_{\alpha\beta}$ refers to a non-zero element of the matrix $\underline{\underline{D}}^{(1)}$. The case of intersite interactions, as, e.g., relevant for the description of d-wave superconductivity, requires a corresponding modification of the following discussion.

However, in the present case, the elements $D^{(1)}_{\alpha\beta}$ are related to the diagonal elements of the density matrix, which we obtain by back-transforming Equation (17):

$$
\begin{aligned}
R^{(1)}_{\alpha\beta} &= T_{\alpha n}\tilde{\chi}^{(0)}_{nm}(\omega)\tilde{V}_{nm}T^{-1}_{m\beta}f(\omega) - |U|T_{\alpha n}\tilde{\chi}^{(0)}_{nm}(\omega)\tilde{D}^{(1)}_{nm}T^{-1}_{m\beta} \\
&\equiv -\tau^y_{\alpha\nu}D^{(1),\dagger}_{\nu\mu}\tau^y_{\mu\beta} = -\tau^y_{\beta\nu}D^{(1)}_{\nu\mu}\tau^y_{\mu\alpha}
\end{aligned} \tag{18}
$$

where we used the following identity for the diagonal elements of the density matrix:

$$
\underline{\underline{R}}^{(1)}_D = -\underline{\underline{\tau}}_y\underline{\underline{D}}^{(1),\dagger}\underline{\underline{\tau}}_y \tag{19}
$$

with

$$
\underline{\underline{\tau}}_y = \begin{pmatrix} \underline{\underline{0}} & -i\underline{\underline{1}} \\ i\underline{\underline{1}} & \underline{\underline{0}} \end{pmatrix} \tag{20}
$$

Equation (18) can be solved for $D_{\nu\mu}^{(1)}$ as

$$
\begin{aligned}
D_{\nu\mu}^{(1)} &= -\tau_{\mu\alpha}^{y} T_{\alpha n} \tilde{\chi}_{nm}^{(0)}(\omega) \tilde{V}_{nm} T_{m\beta}^{-1} \tau_{\beta\nu}^{y} f(\omega) + |U| \tau_{\mu\alpha}^{y} T_{\alpha n} \tilde{\chi}_{nm}^{(0)}(\omega) \tilde{D}_{nm}^{(1)} T_{m\beta}^{-1} \tau_{\beta\nu}^{y} \\
&= -\tau_{\mu\alpha}^{y} T_{\alpha n} \tilde{\chi}_{nm}^{(0)}(\omega) \tilde{V}_{nm} T_{m\beta}^{-1} \tau_{\beta\nu}^{y} f(\omega) + |U| \tau_{\mu\alpha}^{y} T_{\alpha n} \tilde{\chi}_{nm}^{(0)}(\omega) T_{n\rho}^{-1} D_{\rho\sigma}^{(1)} T_{\sigma m} T_{m\beta}^{-1} \tau_{\beta\nu}^{y}
\end{aligned}
$$

where, in the last step, we transformed $\tilde{D}^{(1)}$ back into the original site representation.

We now define

$$
K_{\nu\mu} = -\tau_{\mu\alpha}^{y} T_{\alpha n} \tilde{\chi}_{nm}^{(0)}(\omega) \tilde{V}_{nm} T_{m\beta}^{-1} \tau_{\beta\nu}^{y} \tag{21}
$$

$$
W_{\nu\mu,\rho\sigma} = \tau_{\mu\alpha}^{y} T_{\alpha n} \tilde{\chi}_{nm}^{(0)}(\omega) T_{n\rho}^{-1} T_{\sigma m} T_{m\beta}^{-1} \tau_{\beta\nu}^{y} \tag{22}
$$

so that the equation for $\underline{\underline{D}}^{(1)}$ is given by

$$
D_{\nu\mu}^{(1)} = K_{\nu\mu}(\omega) f(\omega) + |U| W_{\nu\mu,\rho\sigma} D_{\rho\sigma}^{(1)} \tag{23}
$$

or

$$
\underbrace{\left[\delta_{\nu\mu,\rho\sigma} - |U| W_{\nu\mu,\rho\sigma}\right]}_{M_{\nu\mu,\rho\sigma}} D_{\rho\sigma}^{(1)} = K_{\nu\mu}(\omega) f(\omega) \tag{24}
$$

and therefore,

$$
\underline{\underline{D}}^{(1)}(\omega) = \underline{\underline{M}}^{-1}(\omega) \underline{\underline{K}}(\omega) f(\omega). \tag{25}
$$

Inserting the transformed solution of Equation (25) into Equation (17) yields the transformed solution for the density matrix.

Figure 1 shows the magnitude of the first harmonic response for a particular disorder configuration ($V/t = 1$) on a $8 \times 8$ square lattice. We compared the current, obtained from the direct time integration of Equation (6), with the result from Equation (17). For the BCS result, we neglected the time evolution of local charge densities and anomalous correlations in the BdG Hamiltonian Equation (4). This amounts to neglecting the contribution of $\tilde{D}$ in Equation (17), which instead is relevant for the inclusion of collective modes within the RPA. In particular, the phase modes are responsible for the excitations (peaky structures in Figure 1b,d) below the optical gap $2\Delta$; cf. [19]. Note that Figure 1 reports the magnitude of the first-order current response so that the finite BCS response below $2\Delta$ is due to the real part of the current–current correlations. Obviously, the energy resolution in the direct time integration (blue dotted lines) depends on the time interval over which the time integration is performed. In the expansion approach, Equation (17), this resolution can be mimicked by using different values for the parameter $\epsilon$, which shifts the energy into the complex plane. However, a finite $\epsilon$ describes an exponential damping of the time-dependent density matrix over a time scale $\sim 1/\epsilon$. On the other hand, there is no damping in the time integration method, but the integration is simply performed over a fixed time interval. In Figure 1, we use 10 (Panels a, b) and 50 (Panels c, d) periods of the applied vector potential as the time interval for the integration. Note that, for each frequency point, a separate time integration is required.

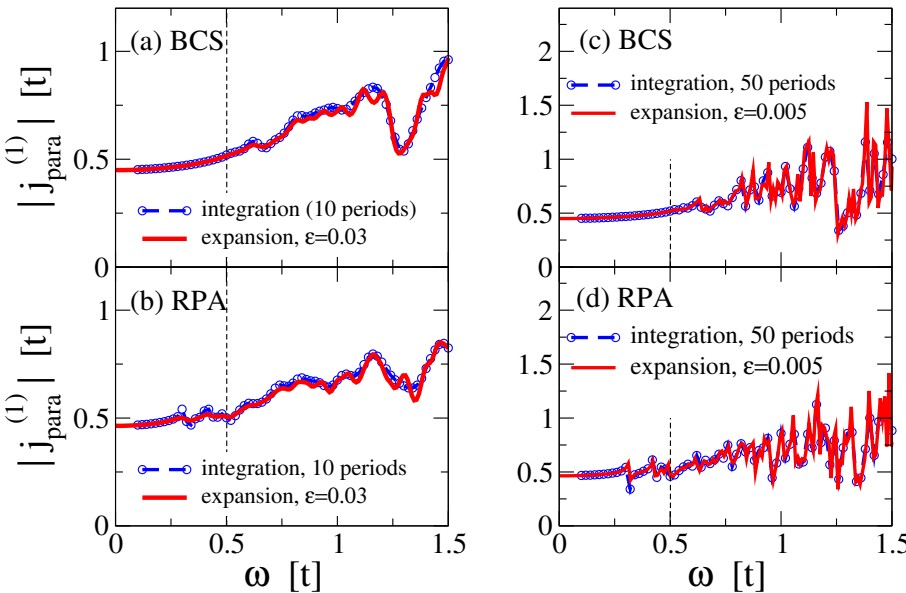

**Figure 1.** Magnitude of the first harmonic response for BCS (**a**,**c**) and RPA (**b**,**d**). The paramagnetic current obtained from the direct time integration Equation (6) is shown by the blue dotted line, and integration was performed over 10 (**a**,**b**) and 50 (**c**,**d**) periods of the applied vector potential. The current evaluated from the expansion Equations (17) and (25) is shown in red. Here, we used $\varepsilon = 0.03t$ (**a**,**b**) and $\varepsilon = 0.005t$ (**c**,**d**) in order to shift the energy into the complex plane, $\omega \to \omega + i\varepsilon$. The vertical dashed line marks the optical gap $2\Delta$. Further parameters: $8 \times 8$ lattice with 56 electrons. $t'/t = 0$; $V_0/t = 1$.

### 3.2. Second Order

We proceed by evaluating the diamagnetic contribution to the third harmonic current $A_0 j_{dia}^{x,(2)}$; cf. Equation (9). Collecting all terms $\sim A_0^2$, we find for the correction to the density matrix in second order:

$$i\underline{\dot{\underline{R}}}^{(2)}(t) = \left[\underline{\underline{R}}^{(2)}(t), \underline{\underline{H}}^{(0)}\right] + \left[\underline{\underline{R}}^{(1)}(t), \underline{\underline{V}}\right]f(t) + \frac{1}{2}\left[\underline{\underline{R}}^{(0)}, \underline{\underline{C}}\right]f^2(t) \tag{26}$$
$$- |U|\left[\underline{\underline{R}}^{(1)}, \underline{\underline{D}}^{(1)}\right] - |U|\left[\underline{\underline{R}}^{(0)}, \underline{\underline{D}}^{(2)}\right]$$

where we defined the matrix:

$$\underline{\underline{C}} = \begin{pmatrix} \underline{c} & \underline{0} \\ \underline{0} & -\underline{c} \end{pmatrix} \tag{27}$$

and

$$c_{nm} = t\left[\delta_{m,n+x} + \delta_{m,n-x}\right] + t'\left[\delta_{m,n+x+y} + \delta_{m,n-x-y}\right] + t'\left[\delta_{m,n+x-y} + \delta_{m,n-x+y}\right].$$

The Fourier transformation yields

$$\omega\underline{\underline{R}}^{(2)}(\omega) = \left[\underline{\underline{R}}^{(2)}(\omega), \underline{\underline{H}}^{(0)}\right] + \int d\nu \left[\underline{\underline{R}}^{(1)}(\nu), \underline{\underline{V}}\right]f(\omega - \nu)$$
$$+ \frac{1}{2}\left[\underline{\underline{R}}^{(0)}, \underline{\underline{C}}\right]\int d\nu f(\nu)f(\omega - \nu) - |U|\int d\nu \left[\underline{\underline{R}}^{(1)}(\nu), \underline{\underline{D}}^{(1)}(\omega - \nu)\right]$$
$$- |U|\left[\underline{\underline{R}}^{(0)}, \underline{\underline{D}}^{(2)}(\omega)\right] \tag{28}$$

which, upon applying the transformation to diagonal states, can be written as

$$\tilde{R}_{nm}^{(2)}(\omega) = \frac{1}{2}\tilde{\chi}_{nm}^{(0)}(\omega)\tilde{C}_{nm}\int d\nu f(\nu)f(\omega - \nu) - |U|\tilde{\chi}_{nm}^{(0)}(\omega)\tilde{D}_{nm}^{(2)}(\omega)$$
$$+ \frac{1}{\omega + E_{nn} - E_{mm}}\int d\nu \left[\tilde{r}^{(1)}(\nu), \tilde{\vartheta}(\omega - \nu)\right]_{nm} f(\nu)f(\omega - \nu). \tag{29}$$

Here, we defined

$$
\begin{aligned}
\tilde{R}_{nm}^{(1)}(\omega) &\equiv \tilde{r}_{nm}^{(1)}(\omega)f(\omega) = \tilde{\chi}_{nm}^{(0)}(\omega)\vartheta_{nm}(\omega)f(\omega) \\
\vartheta_{nm}(\omega) &\equiv \tilde{V}_{nm} - |U|\left[\tilde{d}^{(1)}(\omega)\right]_{nm} \\
\tilde{D}_{nm}^{(1)}(\omega) &\equiv \tilde{d}_{nm}^{(1)}(\omega)f(\omega)\,.
\end{aligned}
$$

We can now follow the same procedure as in the case of the first-order RPA calculation. By transforming to the real space representation, where $D_{nm}^{(2)}$ is again diagonal (similar to Equation (15)), one obtains

$$
\begin{aligned}
D_{\rho\sigma}^{(2)}(\omega) = {} & M_{\rho\sigma,\nu\mu}^{-1}(\omega)G_{\nu\mu}(\omega)\int d\nu\, f(\nu)f(\omega-\nu) \\
& - M_{\rho\sigma,\nu\mu}^{-1}(\omega)\tau_{\mu\alpha}^{y}T_{\alpha,n}\frac{1}{\omega+E_{nn}-E_{mm}}\int d\nu\left[\tilde{r}^{(1)}(\nu),\tilde{\vartheta}(\omega-\nu)\right]_{nm}T_{m\beta}^{-1}\tau_{\beta\nu}^{y}f(\nu)f(\omega-\nu)
\end{aligned}
\tag{30}
$$

where the matrix $\underline{\underline{M}}$ is the same as in Equation (24), and we defined

$$
G_{\nu\mu} = -\frac{1}{2}\tau_{\mu\alpha}^{y}T_{\alpha n}\tilde{\chi}_{nm}^{(0)}(\omega)\tilde{C}_{nm}T_{m\beta}^{-1}\tau_{\beta\nu}^{y}\,.
\tag{31}
$$

Then, by solving Equation (30) and plugging the transformed result into Equation (29), one obtains the second-order frequency-dependent contribution to the density matrix in response to an external field $f(\omega)$.

We exemplify the result for a harmonic external field with $f(\omega) = \delta(\omega - \Omega) + \delta(\omega + \Omega)$. Then, from Equation (9) it turns out that the diamagnetic contribution to $j_x^{(3)}(t)$ is given by $f(t)j_{dia}^{x,(2)}(t)$, which, upon Fourier transformation, implies that $j_x^{(3)}(\omega)$ is given by $j_{dia}^{x,(2)}(\omega - \Omega)$. Thus, the diamagnetic response at $\omega = 3\Omega$ is determined by the density matrix $\tilde{R}_{nm}^{(2)}(\omega - \Omega)$. From Equations (29) and (30), one finds

$$
\begin{aligned}
\tilde{R}_{nm}^{(2)}(\omega-\Omega) = {} & \frac{1}{2}\tilde{\chi}_{nm}^{(0)}(2\Omega)\tilde{C}_{nm}\delta(\omega-3\Omega) - |U|\tilde{\chi}_{nm}^{(0)}(2\Omega)\tilde{D}_{nm}^{(2)}(\omega-\Omega) \\
& + \frac{1}{2\Omega+E_{nn}-E_{mm}}\left[\tilde{r}^{(1)}(\Omega),\tilde{\vartheta}(\Omega)\right]_{nm}\delta(\omega-3\Omega)\,,
\end{aligned}
\tag{32}
$$

with

$$
\begin{aligned}
D_{\rho\sigma}^{(2)}(\omega-\Omega) = {} & M_{\rho\sigma,\nu\mu}^{-1}(2\Omega)G_{\nu\mu}(2\Omega)\delta(\omega-3\Omega) \\
& - M_{\rho\sigma,\nu\mu}^{-1}(2\Omega)\tau_{\mu\alpha}^{y}T_{\alpha,n}\frac{1}{2\Omega+E_{nn}-E_{mm}} \\
& \times \left[\tilde{r}^{(1)}(\Omega),\tilde{\vartheta}(\Omega)\right]_{nm}T_{m\beta}^{-1}\tau_{\beta\nu}^{y}\delta(\omega-3\Omega)\,.
\end{aligned}
\tag{33}
$$

Figure 2 compares the second harmonic response from the direct time integration of Equation (6) with the expansion from Equations (32) and (33) for a particular disorder realisation. As discussed in [19], disorder washes out the resonance at $\omega = \Delta$, and collective modes only slightly increase the intensity of the diamagnetic response. One can also observe that a single parameter $\epsilon$ allows adjusting the response, evaluated from the expansion (red line) to the time-integrated result (blue dotted line) at low energy; however, the agreement in intensity is lost at larger values of $\omega$. For larger time integration intervals (cf., Panels c, d), the agreement is pushed to higher energies.

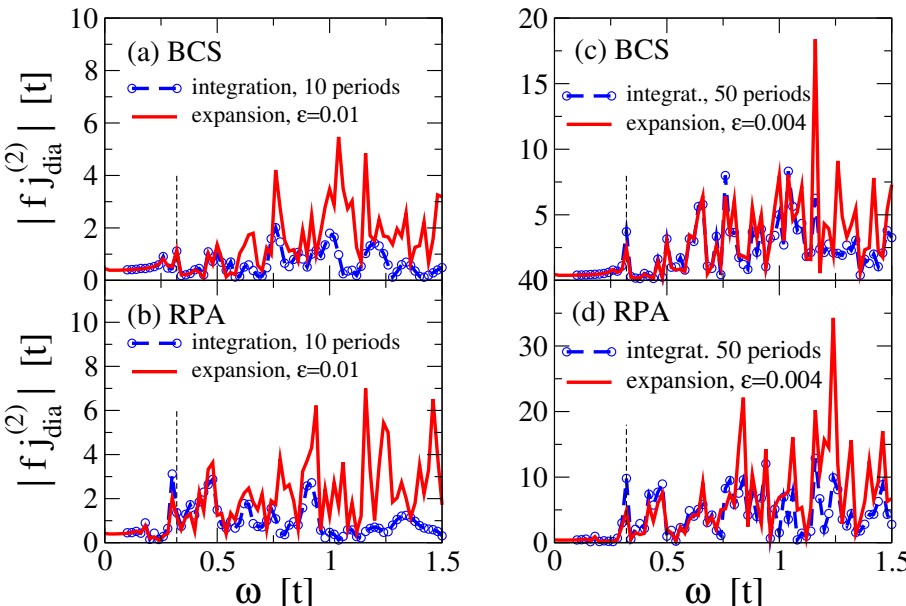

**Figure 2.** Magnitude of the second harmonic response (Fourier transform of $f(f)j_{dia}^2(t)$) for BCS (**a**,**c**) and RPA (**b**,**d**). The diamagnetic current obtained from the direct time integration Equation (6) is shown by the blue dotted line, and integration was performed over 10 (**a**,**b**) and 50 (**c**,**d**) periods of the applied vector potential. The current evaluated from the expansion Equations (32) and (33) is shown in red. Here, we used $\varepsilon = 0.01t$ (**a**,**b**) and $\varepsilon = 0.004t$ (**c**,**d**) in order to shift the energy into the complex plane, $\omega \rightarrow \omega + i\varepsilon$. The vertical dashed line marks the energy at $\Delta$. Further parameters: $8 \times 8$ lattice with 56 electrons. $t'/t = 0$; $V_0/t = 1$.

### *3.3. Third Order*

Finally, we evaluated the paramagnetic contribution to the third harmonic current $j_{para}^{x,(3)}$. Collecting all terms $\sim A_0^3$ in the equation of motion, Equation (6), results in the following equation for the third-order correction to the density matrix

$$
\begin{aligned}
i\underline{\dot{R}}^{(3)}(t) &= \left[\underline{R}^{(3)}(t), \underline{H}^{(0)}\right] + \left[\underline{R}^{(2)}(t), \underline{V}\right]f(t) + \frac{1}{2}\left[\underline{R}^{(1)}, \underline{C}\right]f^2(t) \\
&\quad - \frac{1}{6}\left[\underline{R}^{(0)}(t), \underline{V}\right]f^3(t) - |U|\left[\underline{R}^{(0)}, \underline{D}^{(3)}\right] - |U|\left[\underline{R}^{(1)}, \underline{D}^{(2)}\right] \\
&\quad - |U|\left[\underline{R}^{(2)}, \underline{D}^{(1)}\right].
\end{aligned}
\tag{34}
$$

The Fourier transformation yields

$$
\begin{aligned}
\omega\underline{R}^3(\omega) &= \left[\underline{R}^{(3)}(\omega), \underline{H}^{(0)}\right] + \int d\omega_1 \left[\underline{R}^{(2)}(\omega_1), \underline{V}\right]f(\omega - \omega_1) \\
&\quad + \frac{1}{2}\int d\omega_1 \int d\omega_2 \left[\underline{R}^{(1)}(\omega_1), \underline{C}\right]f(\omega_2)f(\omega - \omega_1 - \omega_2) \\
&\quad - \frac{1}{6}\left[\underline{R}^{(0)}, \underline{V}\right]\int d\omega_1 \int d\omega_2 f(\omega_1)f(\omega_2)f(\omega - \omega_1 - \omega_2) \\
&\quad - |U|\left[\underline{R}^{(0)}, \underline{D}^{(3)}(\omega)\right] - |U|\int d\omega_1 \left[\underline{R}^{(1)}(\omega_1), \underline{D}^{(2)}(\omega - \omega_1)\right] \\
&\quad - |U|\int d\omega_1 \left[\underline{R}^{(2)}(\omega_1), \underline{D}^{(1)}(\omega - \omega_1)\right].
\end{aligned}
\tag{35}
$$

Defining

$$
\underline{D}^{(2)}(\omega) \equiv \int d\nu \underline{d}^{(2)}(\omega, \nu)f(\nu)f(\omega - \nu)
\tag{36}
$$

$$
\underline{R}^{(2)}(\omega) \equiv \int d\nu \underline{r}^{(2)}(\omega, \nu)f(\nu)f(\omega - \nu)
\tag{37}
$$

and transforming to diagonal states, Equation (35) becomes

$$\tilde{R}_{nm}^{(3)}(\omega) = -\frac{1}{6}\tilde{\chi}^{(0)}(\omega)\tilde{V}_{nm}\int d\omega_1\int d\omega_2 f(\omega_1)f(\omega_2)f(\omega-\omega_1-\omega_2)$$

$$+ \quad \frac{1}{\omega+E_{nn}-E_{mm}}\int d\omega_1\int d\omega_2\left[\tilde{r}^{(2)}(\omega_1+\omega_2,\omega_2),\underbrace{\tilde{V}-|U|\tilde{d}^{(1)}(\omega-\omega_1-\omega_2)}_{=\tilde{\vartheta}(\omega-\omega_1-\omega_2)}\right]_{nm}$$

$$\times \quad f(\omega_1)f(\omega_2)f(\omega-\omega_1-\omega_2)$$

$$+ \quad \frac{1}{\omega+E_{nn}-E_{mm}}\int d\omega_1\int d\omega_2\left[\tilde{r}^{(1)}(\omega_1),\frac{1}{2}\tilde{C}-|U|\tilde{d}^{(2)}(\omega-\omega_1,\omega_2)\right]_{nm}$$

$$\times \quad f(\omega_1)f(\omega_2)f(\omega-\omega_1-\omega_2)-|U|\tilde{\chi}_{nm}^{(0)}(\omega)\tilde{D}_{nm}^{(3)}(\omega)\,. \tag{38}$$

Now, we follow the usual procedure and write Equation (38) in terms of the diagonal elements, i.e., $D_{\nu\rho}^{(3)}$, which yields

$$D_{\rho\sigma}^{(3)}(\omega) = -\frac{1}{6}M_{\rho\sigma,\nu\rho}^{-1}K_{\nu\rho}(\omega)\int d\omega_1\int d\omega_2 f(\omega_1)f(\omega_2)f(\omega-\omega_1-\omega_2)$$

$$- \quad M_{\rho\sigma,\nu\rho}^{-1}(\omega)\tau_{\mu\alpha}^y T_{\alpha,n}\frac{1}{\omega+E_{nn}-E_{mm}}\int d\omega_1\int d\omega_2\left[\tilde{r}^{(2)}(\omega_1+\omega_2,\omega_2),\tilde{\vartheta}(\omega-\omega_1-\omega_2)\right]_{nm}$$

$$\times \quad T_{m\beta}^{-1}\tau_{\beta\nu}^y f(\omega_1)f(\omega_2)f(\omega-\omega_1-\omega_2)$$

$$- \quad M_{\rho\sigma,\nu\rho}^{-1}(\omega)\tau_{\mu\alpha}^y T_{\alpha,n}\frac{1}{\omega+E_{nn}-E_{mm}}\int d\omega_1\int d\omega_2\left[\tilde{r}^{(1)}(\omega_1),\frac{1}{2}\tilde{C}-|U|\tilde{d}^{(2)}(\omega-\omega_1,\omega_2)\right]_{nm}$$

$$\times \quad T_{m\beta}^{-1}\tau_{\beta\nu}^y f(\omega_1)f(\omega_2)f(\omega-\omega_1-\omega_2)\,. \tag{39}$$

which, upon inserting into Equation (38), yields the third-order correction to the density matrix. We considered again a harmonic external field with $f(\omega)=\delta(\omega-\Omega)+\delta(\omega+\Omega)$. The contribution of $\tilde{R}_{nm}^{(3)}(\omega)\sim\delta(\omega-3\Omega)$ is then given by

$$\tilde{R}_{nm}^{(3)}(\omega) = -\frac{1}{6}\tilde{\chi}^{(0)}(3\Omega)\tilde{V}_{nm}\delta(\omega-3\Omega)+\frac{1}{3\Omega+E_{nn}-E_{mm}}\left[\tilde{r}^{(2)}(2\Omega,\Omega),\vartheta(\Omega)\right]_{nm}\delta(\omega-3\Omega)$$

$$+ \quad \frac{1}{3\Omega+E_{nn}-E_{mm}}\left[\tilde{r}^{(1)}(\Omega_1),\frac{1}{2}\tilde{C}-|U|\tilde{d}^{(2)}(2\Omega,\Omega)\right]_{nm}\delta(\omega-3\Omega)$$

$$- \quad |U|\tilde{\chi}_{nm}^{(0)}(3\Omega)\tilde{d}_{nm}^{(3)}(3\Omega)\delta(\omega-3\Omega)\,. \tag{40}$$

with

$$\tilde{r}_{nm}^{(2)}(2\Omega,\Omega) = \frac{1}{2}\tilde{\chi}_{nm}^{(0)}(2\Omega)\tilde{C}_{nm}+\frac{1}{2\Omega+E_{nn}-E_{mm}}\left[\tilde{r}^{(1)}(\Omega),\tilde{\vartheta}(\Omega)\right]_{nm}$$

$$- \quad |U|\tilde{\chi}_{nm}^{(0)}(2\Omega)\tilde{d}_{nm}^{(2)}(2\Omega,\Omega)\,. \tag{41}$$

$$\tilde{d}_{nm}^{(2)}(2\Omega,\Omega) = M_{\rho\sigma,\nu\mu}^{-1}(2\Omega)G_{\nu\mu}(2\Omega)$$

$$- \quad M_{\rho\sigma,\nu\mu}^{-1}(2\Omega)\tau_{\mu\alpha}^y T_{\alpha,n}\frac{1}{2\Omega+E_{nn}-E_{mm}}\left[\tilde{r}^{(1)}(\Omega),\tilde{\vartheta}(\Omega)\right]_{nm}T_{m\beta}^{-1}\tau_{\beta\nu}^y \tag{42}$$

$$\tilde{d}_{nm}^{(3)}(3\Omega) = -\frac{1}{6}M_{\rho\sigma,\nu\rho}^{-1}(3\Omega)K_{\nu\rho}(3\Omega)$$

$$- \quad M_{\rho\sigma,\nu\rho}^{-1}(3\Omega)\tau_{\mu\alpha}^y T_{\alpha,n}\frac{1}{3\Omega+E_{nn}-E_{mm}}\left[\tilde{r}^{(2)}(2\Omega,\Omega),\tilde{\vartheta}(\Omega)\right]_{nm}T_{m\beta}^{-1}\tau_{\beta\nu}^y$$

$$- \quad M_{\rho\sigma,\nu\rho}^{-1}(3\Omega)\tau_{\mu\alpha}^y T_{\alpha,n}\frac{1}{3\Omega+E_{nn}-E_{mm}}\left[\tilde{r}^{(1)}(\Omega),\frac{1}{2}\tilde{C}-|U|\tilde{d}^{(2)}(2\Omega,\Omega)\right]_{nm}T_{m\beta}^{-1}\tau_{\beta\nu}^y\,. \tag{43}$$

For the same disorder configuration as was used for Figures 1 and 2, we show in Figure 3 the third harmonic response from Equations (17) and (25) as compared to the direct time integration of Equation (6). Consistent with our previous results [19], the strongly disordered ordered sample displays a low paramagnetic energy response at $\omega = \Delta$, both in the BCS and RPA, where the latter is enhanced by the contribution from the collective modes. As in case of the diamagnetic contribution (cf. Figure 2), the "expansion result" (red) for a fixed $\epsilon$ parameter can be adjusted to the time-integrated spectrum (blue dotted line) at low energies, but with a decreasing number of periods in the time integration, the agreement in intensity is lost at higher energies. This is particularly visible in Figure 3d, where the contribution from band excitations leads to significantly larger intensities for the small $\epsilon = 0.004$ as compared to the time integration over 50 periods of the applied vector potential.

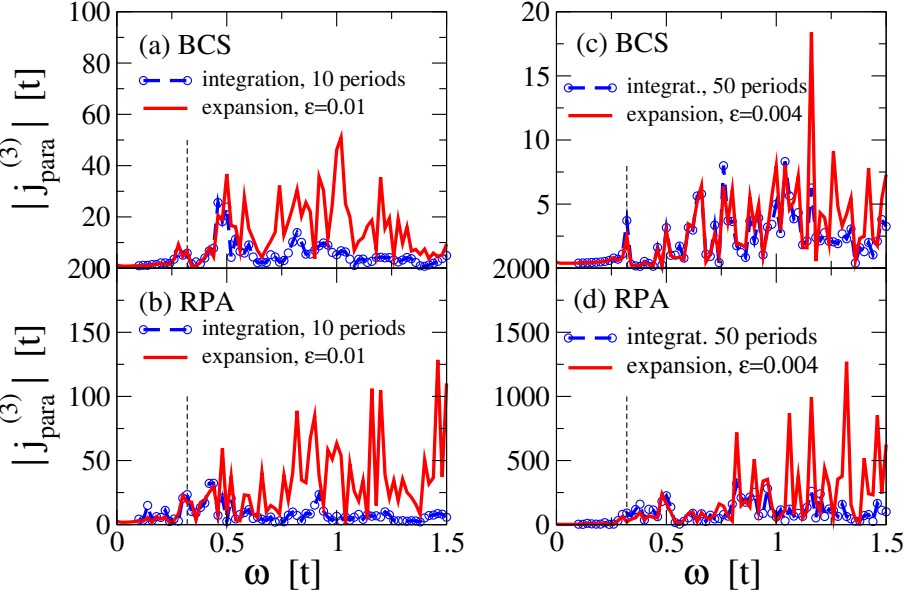

**Figure 3.** Magnitude of the third harmonic response (Fourier transform of $j^3_{para}(t)$) for BCS (**a**,**c**) and RPA (**b**,**d**). The diamagnetic current obtained from the direct time integration Equation (6) is shown by the blue dotted line, and integration was performed over 10 (**a**,**b**) and 50 (**c**,**d**) periods of the applied vector potential. The current evaluated from the expansion Equations (17) and (25) is shown in red. Here, we used $\varepsilon = 0.01t$ (**a**,**b**) and $\varepsilon = 0.004t$ (**c**,**d**) in order to shift the energy into the complex plane, $\omega \to \omega + i\varepsilon$. The vertical dashed line marks the gap $\Delta$. Further parameters: $8 \times 8$ lattice with 56 electrons. $t'/t = 0$; $V_0/t = 1$.

## 4. Conclusions

We presented a detailed comparison of two approaches to evaluate the higher-harmonic-current response to an applied electromagnetic field for disordered and superconducting systems on a lattice. The first method is based on the direct time integration of the equation of motion, Equation (6), as was used in [19] for the investigation of the influence of collective modes in disordered s-wave superconductors. Since, in this case, the higher harmonic contribution has to be extracted numerically from the total response, the calculation has to be performed for at least three different magnitudes of the vector potential for each frequency. Together with the fact that, in order to obtain a reasonable frequency resolution, the integration has to be performed over a significant number of periods of the applied vector potential, this method is limited to a small number of lattice sites. On the other hand, it is rather flexible with regard to the simulation of different pump–probe protocols, which can be easily implemented in the formalism.

Alternatively, one can compute the THG from an expansion of the density matrix in powers of the applied vector potential. As we demonstrated in the present paper, the equations of motion for the individual components can be directly solved in the frequency

space from which the currents in the various orders are obtained. In [26], this approach was applied to the evaluation of the third harmonic response in d-wave superconductors, where, at least in the BCS limit, one could treat much larger systems than via the direct time integration of the density matrix. In this paper, we showed how RPA corrections can be included in the formalism. An open issue is the problem of how these RPA corrections can be separated into contributions from the amplitude, phase, and charge modes, which, on the other hand, can be easily accomplished within the time integration method.

**Funding:** This research was funded by the Deutsche Forschungsgemeinschaft under SE806/20-1.

**Data Availability Statement:** Data is contained within the article

**Acknowledgments:** The author is deeply indebted to Lara Benfatto and Claudio Castellani for the stimulating discussions on this topic.

**Conflicts of Interest:** The author declares no conflict of interest.

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
