# Peer review of "On the Evaluation of Higher-Harmonic-Current Responses for High-Field Spectroscopies in Disordered Superconductors"

_condensedmatter, doi:10.3390/condmat8040095_

Round 1

Reviewer 1 Report

Comments and Suggestions for Authors

The presented theoretical study is dedicated on non-linear (third order) optical response of disordered superconductors and the corresponding superconducting order parameter dynamics in the frames of tight biding models - unconventional superconductivity mechanism.

The study is comprehensively developed and will contribute to the field with expanding of fundamental level and functionality of THz optical methods for deeper understanding of electronic structure of superconductors. 

I would suggest to the author to specify the tittle in the context of optical conductivity "On the evaluation of higher harmonic optical conductivity in disordered superconductors" - for instance, considering that there are also studies on non-linear AC magnetic susceptibility and AC electro-transport current which are aimed on vortex matter dynamics and practical directions of AC losses in superconducting cables. 

With respect.   

Author Response

I would like to thank the referee for her(his) positive evaluation of the

manuscript. According to his(her) suggestion I have modified the title to

"On the evaluation of higher harmonic current responses for high field spectroscopies in disordered superconductors".

Reviewer 2 Report

Comments and Suggestions for Authors

In the presented work entitled „On the evaluation of higher harmonic current responses in disordered superconductors” by G. Seibold, the Author discuss a formalism which allows for the calculation of a higher harmonic current response to an applied electric field for disordered superconducting systems. This allows the evaluation of higher order response functions on significantly larger lattices than one can achieve with a previously used approach, which is based on a direct temporal integration of the equation of motion for the complete density matrix. In case of small lattices, where both methods can be applied by also including the contribution of collective modes, demonstrate the agreement of the corresponding results is demonstrated.

In general, the article is written in a well-organized manner, and I did not notice any major errors in the conducted analysis. The discussion is clear and allows readers to get all the technical aspects of the presented investigations. The subject of the manuscript also appears to be timely although it does not present major breakthrough. Still in my opinion the presented results are worth publishing and may be of interest to the readers. I have only one minor comment, the Author call the employed models, the tight-binding models. While this is partially true (the non-interacting part is indeed the so-called tight-binding part) the entire model should be called simply the Hubbard model in my opinion (in fact the Author uses this name in the text also). Other than that, I recommend the presented paper for publication.

(* end of report *)

Author Response

I would like to thank the referee for her(his) positive evaluation of the manuscript. According to her(his) suggestion I have specified in the text that the tight-binding models include also local or intersite interactions.